# The asymmetric expression of HSPA2 in blastomeres governs the first embryonic cell-fate decision

Jiayin Gao[1,2,3,4,5,6,7,8†], Jiawei Wang[1,2,3,4,5,6,7,8†], Shiyu Liu[1,2,3,4,5,6,7,8], Jinzhu Song[1,2,3,4,5,6,7,8], Chuanxin Zhang[1,2,3,4,5,6,7,8], Boyang Liu[1,2,3,4,5,6,7,8*], Keliang Wu[1,2,3,4,5,6,7,8*]

[1]Institute of Women, Children and Reproductive Health, Shandong University, Jinan, China; [2]State Key Laboratory of Reproductive Medicine and Offspring Health, Shandong University, Jinan, China; [3]National Research Center for Assisted Reproductive Technology and Reproductive Genetics, Shandong University, Jinan, China; [4]Key Laboratory of Reproductive Endocrinology (Shandong University), Ministry of Education, Jinan, China; [5]Shandong Technology Innovation Center for Reproductive Health, Jinan, China; [6]Shandong Provincial Clinical Research Center for Reproductive Health, Jinan, China; [7]Shandong Key Laboratory of Reproductive Medicine, Shandong Provincial Hospital Affiliated to Shandong First Medical University, Jinan, China; [8]Research Unit of Gametogenesis and Health of ART-Offspring, Chinese Academy of Medical Sciences, Jinan, China

*For correspondence:
byliu@sdu.edu.cn (BL);
wukeliang@sdu.edu.cn (KW)

†These authors contributed equally to this work

Competing interest: The authors declare that no competing interests exist.

## eLife Assessment

This **useful** study by Gao et al identifies Hspa2 as a heterogeneous transcript in the early embryo and proposes a plausible mechanism showing interactions with Carm1. The authors propose that variability in HSPA2 levels among blastomeres at the 4-cell stage skews their relative contribution to the embryonic lineage. Given only 4 other heterogeneous transcripts/non-coding RNA have been proposed to act similarly at or before the 4-cell stage, this would be a key addition to our understanding of how the first cell fate decision is made. While this is a **solid** study, further data are needed to fully support the conclusions.

**Abstract** The first cell-fate decision is the process by which cells of an embryo take on distinct lineage identities for the first time, thus representing the beginning of developmental patterning. Here, we demonstrate that the molecular chaperone heat shock protein A2 (HSPA2), a member of the 70 kDa heat shock protein (HSP70) family, is asymmetrically expressed in the late 2-cell stage of mouse embryos. The knockdown of *Hspa2* in one of the 2-cell blastomeres prevented its progeny predominantly towards the inner cell mass (ICM) fate. In contrast, the overexpression of *Hspa2* in one of the 2-cell blastomeres did not induce the blastomere to differentiate towards the ICM fate. Furthermore, we demonstrated that HSPA2 interacted with CARM1 and its levels correlated with ICM-associated genes. Collectively, our results identify HSPA2 as a critical early regulator of the first cell-fate decision in mammalian 2-cell embryos.

## Introduction

The preimplantation development of mammalian embryos is a complex and orderly process (*Mu et al., 2022*). The first cell-fate decision is a key event in the preimplantation development in mammalian embryos. Specifically, the first cell-fate decision occurs at the morula stage following embryonic compaction and results in the division of embryonic cells into two distinct lineages: the trophectoderm (TE) and the inner cell mass (ICM). The TE is a single layer surrounding the fluid-filled cavity known as the blastocoel that provides extraembryonic structures such as the placenta, whereas the ICM, which is attached to the inside of the TE, contains pluripotent cells that give rise to the fetus and extraembryonic tissues (*Cockburn and Rossant, 2010*; *Liu et al., 2021*; *Rossant and Tam, 2009*). However, it remains an open question of how these cells shape the first cell-fate decision of preimplantation development, and when they begin to trigger the initial differentiation signals.

The asymmetry of the blastocyst may be traced back to the asymmetric nuclear abundance of coactivator-associated arginine methyltransferase 1 (CARM1) at the 4-cell stage (*Goolam et al., 2016*; *Shi et al., 2015*; *Torres-Padilla et al., 2007*). The heterogeneous activity of CARM1 leads to the differential methylation of H3 methylation of arginine 26 (H3R26me). More extensive methylation of H3R26me can result in a higher increase in the expression levels of ICM-specific genes (e.g. *Pou5f1*, *Nanog* and *Sox2*) and global chromatin accessibility, which direct their progeny into the ICM fate (*Sotomayor-Lugo et al., 2024*). Recent studies have further suggested that the asymmetry of blastomeres can even be traced back to the 2-cell stage. For example, a long non-coding RNA, *Gm45001* (*LincGET*), is known to be asymmetrically expressed in the nuclei of blastomeres in 2- and 4-cell embryos. Specifically, *Gm45001* and CARM1 are known to form a complex that promotes H3R26me and activates the expression of ICM genes (*Wang et al., 2018*). In addition, HMGA1, a member of the high mobility group (HMG) protein family, is also known to exhibit heterogeneity as early as the 2-cell stage and involves in cell fate differentiation (*Wang et al., 2021*). Collectively, these studies suggest that molecular heterogeneity takes place in the early stage of embryonic development and is associated with cell fate differentiation.

Despite these findings, the specific factors that exhibit heterogeneity at the 2-cell stage and the mechanisms by which the cell fate and differentiation can be decided still require further investigation. The molecular chaperone heat shock protein A2 (HSPA2) is a member of the 70 kDa heat shock protein (HSP70) family that is evolutionarily conserved in human and metazoan. It is originally identified in male germ cells and described as a testis-specific protein that is essential for spermatogenesis (*Gómez-Torres et al., 2023*; *Scieglinska and Krawczyk, 2015*). The *Hspa2* gene is located on chromosome 14 (14q24.1) (*Zhang et al., 2013*), and the aberrant expression of HSPA2 in testes was shown to induce primary spermatocytes to arrest in meiosis I and undergo apoptosis, thus leading to male infertility (*Bonnycastle et al., 1994*; *Dix et al., 1996*). HSPA2 is expressed in somatic tissues and during mouse embryogenesis (*Murashov and Wolgemuth, 1996*; *Rupik et al., 2006*), and high levels of HSPA2 expression play a critical role in the genesis and progression of carcinoma (*Garg et al., 2010a*; *Garg et al., 2010b*). However, the role of HSPA2 in lineage differentiation of embryonic cells is yet to be unveiled.

In this study, we separated blastomeres from the 4-cell stage and cultured to the morula stage, analyzed the gene expression pattern by using Smart-seq2 to identify differentially expressed genes. *Hspa2* was identified and was shown to be asymmetrically distributed in 2- to 4-cell blastomeres. The knockdown of *Hspa2* in one of the 2-cell blastomeres prevented its progeny predominantly towards the ICM fate. Furthermore, HSPA2 and CARM1 were found to form a protein complex and regulated ICM-specific gene expression. Collectively, these results suggest that *Hspa2* expression patterns can bias cell-fate in the mouse embryo.

## Results

### *Hspa2* is asymmetrically expressed among late 2- and 4-cell blastomeres

In our previous study, we established a culture system that allows zygotes without their zona pellucida to develop in vitro until the blastocyst stage (*Song et al., 2022*). Previous studies have mainly focused on gene heterogeneity in 4-cell stage mouse embryos (*Goolam et al., 2016*; *Torres-Padilla et al., 2007*). Here, in order to expand the heterogeneity among blastomeres and identify

factors that may involve in cell lineage differentiation, single blastomeres from the mouse 4-cell embryo were separated and cultured to the morula stage. By using Smart-seq2 technology, we analyzed the gene expression pattern of the morulae, which developed from the 4-cell blasto-meres (*Figure 1A*). *Hspa2* was identified by differential gene expression analysis and showed distinct mRNA levels among the four morulae (*Figure 1B*). In addition, the level of *Nanog* and *Hspa2* mRNA showed stronger positive correlation ($R=0.624$) in gene expression than random gene pairs ($R=0.029$), indicating that *Nanog* expression levels increased with the increase of *Hspa2* (*Figure 1C*). When the late 2-cell embryo was divided into single blastomeres and cultured to the morula stage, we obtained similar results that *Hspa2* mRNA distinctly distributed between the two morulae (*Figure 1—figure supplement 1A, B*). Next, we investigated the expression of *Hspa2* during preimplantation embryo development from meiosis II to the blastocyst stage. qRT-PCR analysis showed that *Hspa2* mRNA was stored maternally at the MII oocyte and decreased at the two-cell stage but remarkably increased from the 2-cell to the 4-cell stage, coinciding with zygotic genome activation (ZGA). There was a slight change in expression as embryos developed to the 8-cell and morula stages, and developed into blastocysts that had higher expression levels than the 2-cell stage (*Figure 1D*).

To directly examine the heterogeneity among individual blastomeres, we separated 4-cell embryos into individual four blastomeres which were then analyzed by single-cell qRT-PCR and Jess TM Simple Western System based automated western immunoblotting with high sensitivity. The results showed that the mRNA and protein levels of HSPA2 differed among the four blastomeres (*Figure 1E and F*), and confirmed by fluorescence in situ hybridization (FISH) analysis (*Figure 1G*). We also identi-fied heterogeneous HSPA2 expression during the late 2-cell stage (*Figure 1H–J*). For the early 2-cell stage, however, HSPA2 was symmetrically distributed between the two blastomeres (*Figure 1—figure supplement 1C–E*). These results indicate that the heterogeneity occurred as early as the late 2-cell stage. Thus, we hypothesized that the heterogeneous distribution of HSPA2 in late 2 cell plays a direct or indirect role in the lineage commitment of early embryonic stages.

## The reduced expression of *Hspa2* leads to the reduced expression of ICM-marker

To test our hypothesis, we knocked down the *Hspa2* mRNA level by injecting *Hspa2*-specific small interfering RNA (siRNA) into the cytoplasm of zygote, and NC-FAM (negative control with FAM label) siRNA-injected embryos served as the control (*Figure 2A*). The differentially expressed genes were detected by RNA-seq for *Hspa2* siRNA (Knockdown; KD) and control (NC) groups at the 8-cell stage (*Figure 2B*). In total, 96 up-regulated genes and 410 down-regulated genes were identified in the *Hspa2*-KD group when compared to the NC group (*Figure 2C and D*). To investigate the potential biological functions of the differentially expressed genes (DEGs), we conducted GO and KEGG enrich-ment analysis. The most significant GO enriched terms were closely related to cell fate commitment, stem cell population maintenance, stem cell differentiation and blastocyst development (*Figure 2E*). According to KEGG analysis, the down-regulated genes were primarily associated with Hippo and Wnt signaling pathways (*Figure 2F*), which are known to play an important role in cell proliferation, differentiation, and fate decisions in embryonic development (*de Jaime-Soguero et al., 2018*; *Nish-ioka et al., 2009*; *Świerczek Lasek et al., 2021*; *ten Berge et al., 2011*). Furthermore, the blastocyst formation rate in the *Hspa2*-KD group ($52.3 \pm 2.5\%$) was significantly lower than that in the NC group ($81.6 \pm 2.0\%$; *Figure 2G and H*), thus suggesting that HSPA2 plays an essential role in preimplanta-tion embryonic development. In addition, we found that the down-regulation of *Hspa2* resulted in a remarkable down-regulation of specific ICM-marker genes (*Pou5f1*, *Sox2,* and *Nanog*) at the 4 cell and blastocyst stages. Meanwhile, the mRNA levels of a TE-marker gene, *Cdx2*, did not significantly change between the groups (*Figure 2I and J*). We then used automated western immunoblotting to detect the protein levels of ICM and TE markers in *Hspa2*-KD embryos, and found that the expression levels of ICM-marker proteins (OCT4 and SOX2) were inhibited when HSPA2 was down-regulated at the 4cell and blastocyst stages, while the TE-marker protein CDX2 was not significantly different between the groups (*Figure 2K and L*). Similarly, for mouse embryonic stem cells (mESCs), the knock-down of HSPA2 resulted in a significant reduction for the ICM-marker proteins, and no change for CDX2 (*Figure 2M*). These results suggest that HSPA2 plays a role in the regulation of ICM-specific gene expression.

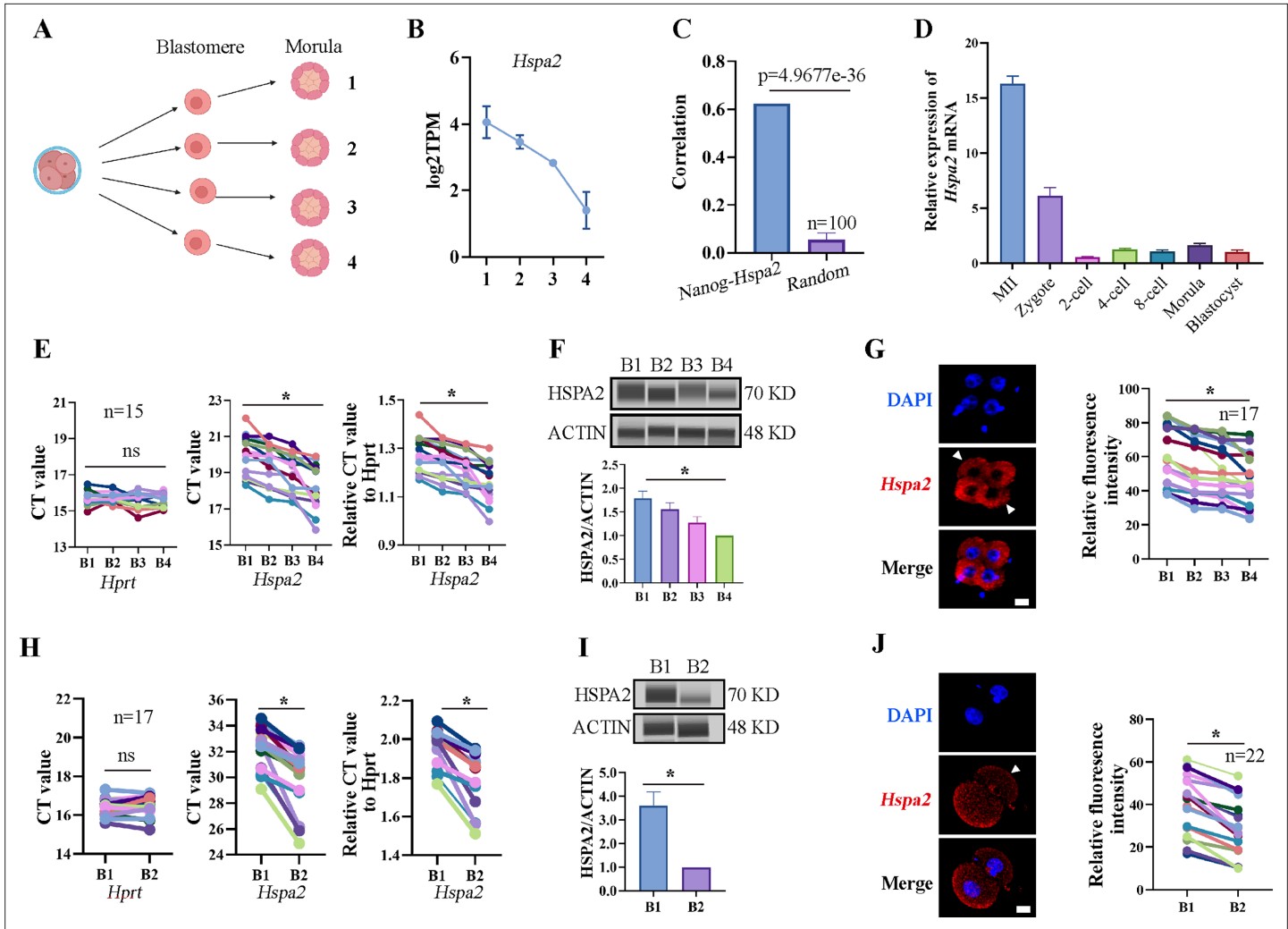

**Figure 1.** *Hspa2* is asymmetrically expressed among mouse 2- and 4-cell blastomeres. (**A**) Schematic overview. The 4-cell embryo divided into single blastomeres and cultured to the morula stage. Created in BioRender. (**B**) *Hspa2* expression is heterogeneous in morula developed from 4-cell blastomeres. Three experimental replicates were performed. (**C**) *Nanog* and *Hspa2* show stronger positive correlation (*R*=0.624) in gene expression than random gene pairs (n=100, n means the number of random gene pairs). (**D**) qPCR analysis of the relative expression of *Hspa2* mRNA from the meiosis II to blastocyst stages. The error bars represent SEM. About 20 embryos of each stage were used and three experimental replicates were performed. (**E, H**) Single-cell qRT-PCR results showing *Hspa2* distribution between 4- (**E**) (n=15), late 2- (**H**) (n=17) cell blastomere. *Hprt* as the housekeeping gene. CT values are used for level analysis. Two-tailed Student's t tests were used for the statistical analysis. Blastomeres are named B1 to B4 (**E**) or B1 and B2 (**H**) according to increasing *Hspa2* concentration. *$P<0.05$. (**F, I**) Automated western immunoblotting results showing HSPA2 distribution between 4- (**F**) and late 2- (**I**) cell blastomeres. Blastomeres are named B1 to B4 (**E**) or B1 and B2 (**H**) according to decreasing *Hspa2* concentration. The band intensity was assessed. There experimental replicates were performed. *$p<0.05$. (**G, J**) FISH of *Hspa2* mRNA in early 4- (**G**) (n=17), late 2- (**J**) (n=22) cell embryos. Blastomeres with low expression as white triangles. RNA-FISH fluorescence quantified and normalized to the nucleus with the strongest staining per individual 2- or 4-cell embryo. Blastomeres are named B1 to B4 (**G**) or B1 and B2 (**J**) according to decreasing *Hspa2* concentration. Three experimental replicates were performed. Bar, 25 μm. *$p<0.05$.

The online version of this article includes the following source data and figure supplement(s) for figure 1:

**Source data 1.** PDF file containing original western blots, indicating relevant band.

**Source data 2.** Original files for western blot analysis.

**Figure supplement 1.** *Hspa2* is asymmetrically expressed between mouse 2- and 4-cell blastomeres.

**Figure supplement 1—source data 1.** PDF file containing original western blots, indicating relevant band.

**Figure supplement 1—source data 2.** Original files for western blot analysis.

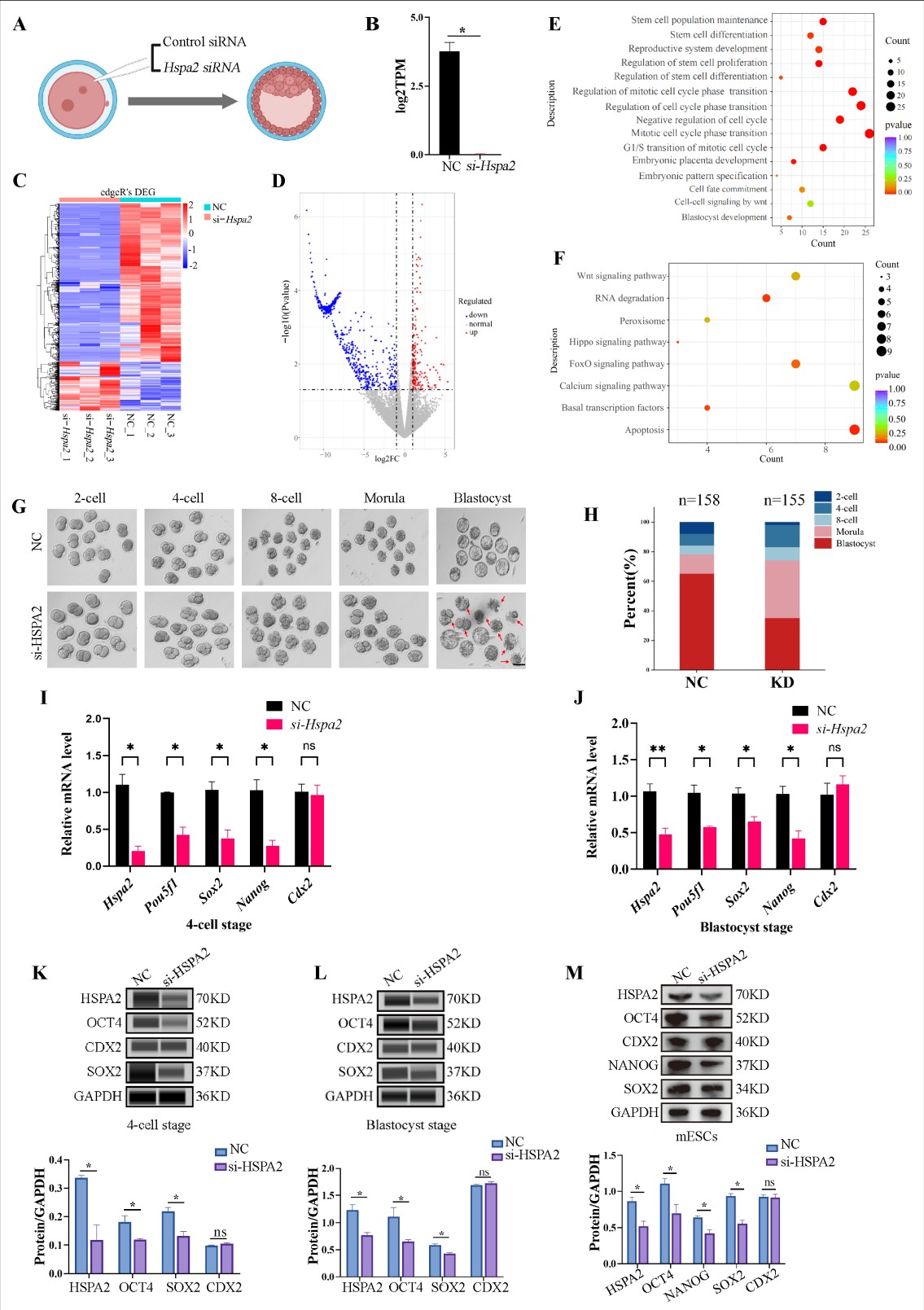

**Figure 2.** Knockdown of *Hspa2* leads to decreased expression of ICM-mark at both mRNA and protein level. (**A**) Schematic diagram of microinjection of zygotic embryos. Created in BioRender. (**B**) Bar charts showing the expression of *Hspa2* in the negative control (NC) group (injected with negative control siRNA) and siRNA targeting group based on RNA-seq. The error bars denote the standard deviations of three experimental replicates of RNA-seq. *p<0.05. (**C**) Heat map of the hierarchical clustering analysis of the DEGs between NC and KD groups. (**D**) Volcano plots of significantly differentially

*Figure 2 continued on next page*

*Figure 2 continued*

expressed genes (DEGs). Red, up-regulated; Blue, down-regulated. (**E**) GO enrichment analysis from down-regulated DEGs between NC and KD groups. (**F**) Significantly down-regulated DEGs were analyzed by KEGG enrichment. (**G**) Representative images of preimplantation embryos in the NC and KD groups from 2-cell to blastocyst. Red arrowhead denotes degenerating blastocysts. Bar, 100 μm. (**H**) Bar plots showing the developmental rates of NC group and KD group at the blastocyst stage. Three experimental replicates were performed. (**I, J**) qRT-PCR of 4-cell embryo and blastocyst showed that ICM-marker gene (*Pou5f1, Sox2,* and *Nanog*) mRNA was significantly lower in the interference compared with the NC group. There were no significant differences in *Cdx2* between the two groups. *p<0.05, **p<0.01. (**K, L**) Automated western blotting analysis of ICM-marker protein (OCT4, SOX2, and NANOG) expression in the interference and NC groups. Three experimental replicates were performed. *p<0.05. (**M**) Western blot analysis of ICM-marker protein (OCT4, SOX2, and NANOG) and TE-marker protein (CDX2) expression in mESCs. The band intensity was assessed with ImageJ. Three experimental replicates were performed. *p<0.05.

The online version of this article includes the following source data for figure 2:

**Source data 1.** PDF file containing original western blots, indicating relevant band.

**Source data 2.** Original files for western blot analysis.

## The knockdown of *Hspa2* expression prevents blastomeres from an ICM fate

To investigate the functional role of HSPA2 in the first lineage segregation during early embryonic development, we examined the proportion of ICM and TE after *Hspa2*-KD. It was showed that the number of ICM and total cells in the blastocyst were significantly decreased, but the number of TE cells did not change significantly (*Figure 3A and B*). We then randomly co-injected green fluorescent protein (*Gfp*) mRNA with either *Hspa2*-siRNA or NC-FAM into one blastomere of the 2-cell embryos to trace the fate of its descendent cells into the blastocyst (*Figure 3C*). Knock-down *Hspa2* in one blastomere at the 2-cell stage also led to developmental defects (*Figure 3—figure supplement 1A, B*). Compared with the GFP +NC group (48.2% ± 7.4%), the GFP +*Hspa2* KD group (33.8 %±4.3%) exhibited a significantly reduced ratio of progeny cells with an ICM fate. Moreover, the number of ICM-marker OCT4 positive cells was found to be significantly reduced in the GFP +*Hspa2* KD group (27.1 ± 3.8%) than in the GFP +NC group (37.0% ± 2.5%; *Figure 3D and E*). This is consistent with our GO enriched terms of RNA-Seq in that the negative regulation of cell cycle, G1/S transition of mitotic cell cycle, mitotic cell cycle phase transition and regulation of mitotic cell cycle phase transition were enriched in *Hspa2*-KD group (*Figure 2E*). Collectively, these results suggested that HSPA2 has a role in ICM lineage establishment and its deficiency leads to partial defects in the ICM cells.

## The overexpression of *Hspa2* does not induce blastomere cells to bias an ICM fate

To determine whether the redundant level of HSPA2 protein also affected first lineage segregation, we overexpressed *Hspa2* by microinjecting *Hspa2* mRNA (approximately 400 ng/μL) into the zygote. Western blotting assays showed that the HSPA2 was overexpressed in the blastocyst stage (*Figure 4—figure supplement 1A*). However, we observed that the blastocyst formation rate was not affected significantly (*Figure 4A and B*), thus showing that the overexpression of *Hspa2* did not exert adverse effects on embryonic development. In addition, we verified that the mRNA levels of the ICM-marker genes (*Pou5f1, Sox2,* and *Nanog*) and the TE-marker gene *Cdx2* at the 4 cell and blastocyst stage did not differ significantly (*Figure 4C and D*). We also examined the number of ICM and TE cells by immunostaining OCT4 and CDX2 in blastocysts from the overexpression and control groups and found they were not altered significantly (*Figure 4E and F*). To trace the fate of the descendent, we co-injected GFP mRNA along with either *Hspa2* or NC mRNA randomly into one blastomere of 2-cell embryos and counted the proportion of ICM and TE cells of blastocysts. We found that the percentage of GFP cells that were present in the ICM at the blastocyst stage was similar between the two groups (48.2±7.4 vs 47.3±4.4), and the percentage of ICM cells also did not differ significantly (37.0±2.5 vs 36.2±1.9; *Figure 4G and H*). These observations indicated that the redundant level of HSPA2 was not sufficient to induce blastomere cells to bias an ICM fate.

## HSPA2 interacts with CARM1 and alters H3R26me2 levels

To further investigate the mechanism underlying the function of HSPA2 in cell-fate decision during early embryonic development, we next analyzed differentially expressed proteins (DEPs) at the

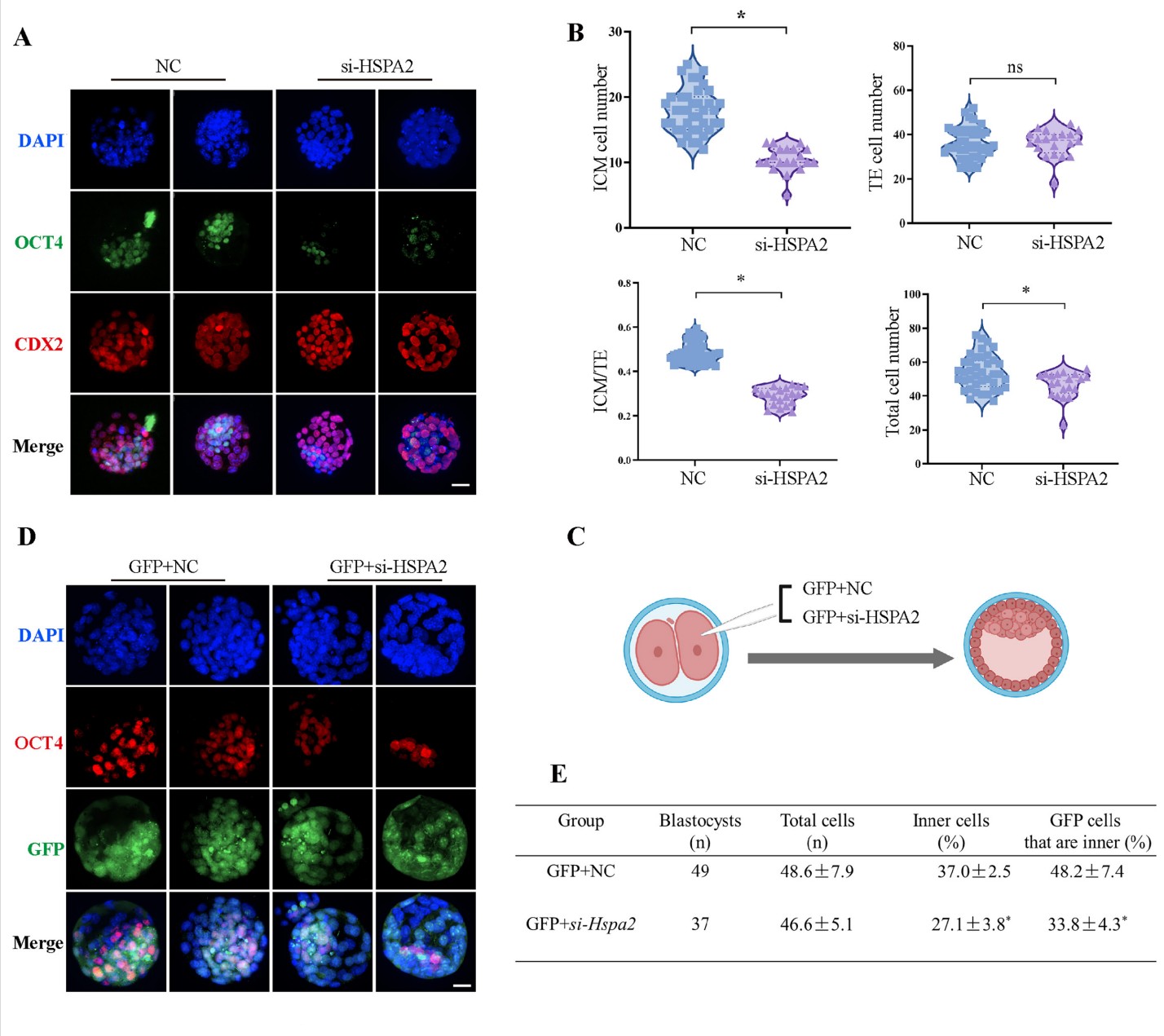

**Figure 3.** Knockdown of *Hspa2* prevents blastomeres from undergoing an ICM fate. (**A**) Immunofluorescence of OCT4 (green) and CDX2 (red) in NC (n=40) and si-*Hspa2* (n=36) groups. OCT4 was used as an ICM marker. Nuclei were visualized with DAPI staining (blue). Bar, 50 µm. (**B**) The number of ICM and TE. OCT4 was used as an ICM marker and CDX2 was used as a TE marker. *p<0.05. (**C**) Schematic diagram of microinjection of early 2-cell embryo single blastomeres. Created in BioRender. (**D**) OCT4 and GFP fluorescent staining of blastocysts. OCT4 was used as an ICM marker. The results show that less OCT4-positive cells are GFP-positive in embryos injected with GFP and *Hspa2 siRNA* (GFP +si-*Hspa2*). Three experimental replicates were performed. Bar, 50 µm. (**E**) Analysis of the distribution of progeny of injected blastomere at the blastocyst stage based on fluorescence staining. Total cells (n) are the total number of cells in the blastocyst; inner cells (%) is the percentage of inner cells relative to the total number of cells in the blastocyst; GFP cells that are inner (%) is the percentage of GFP-positive inner cells relative to the total number of GFP-positive cells in the blastocyst. Data derived from three independent experiments and presented as mean ± standard deviation. Three experimental replicates were performed. *p<0.05.

The online version of this article includes the following figure supplement(s) for figure 3:

**Figure supplement 1.** Knockdown of *Hspa2* prevents blastomeres from undergoing an ICM fate.

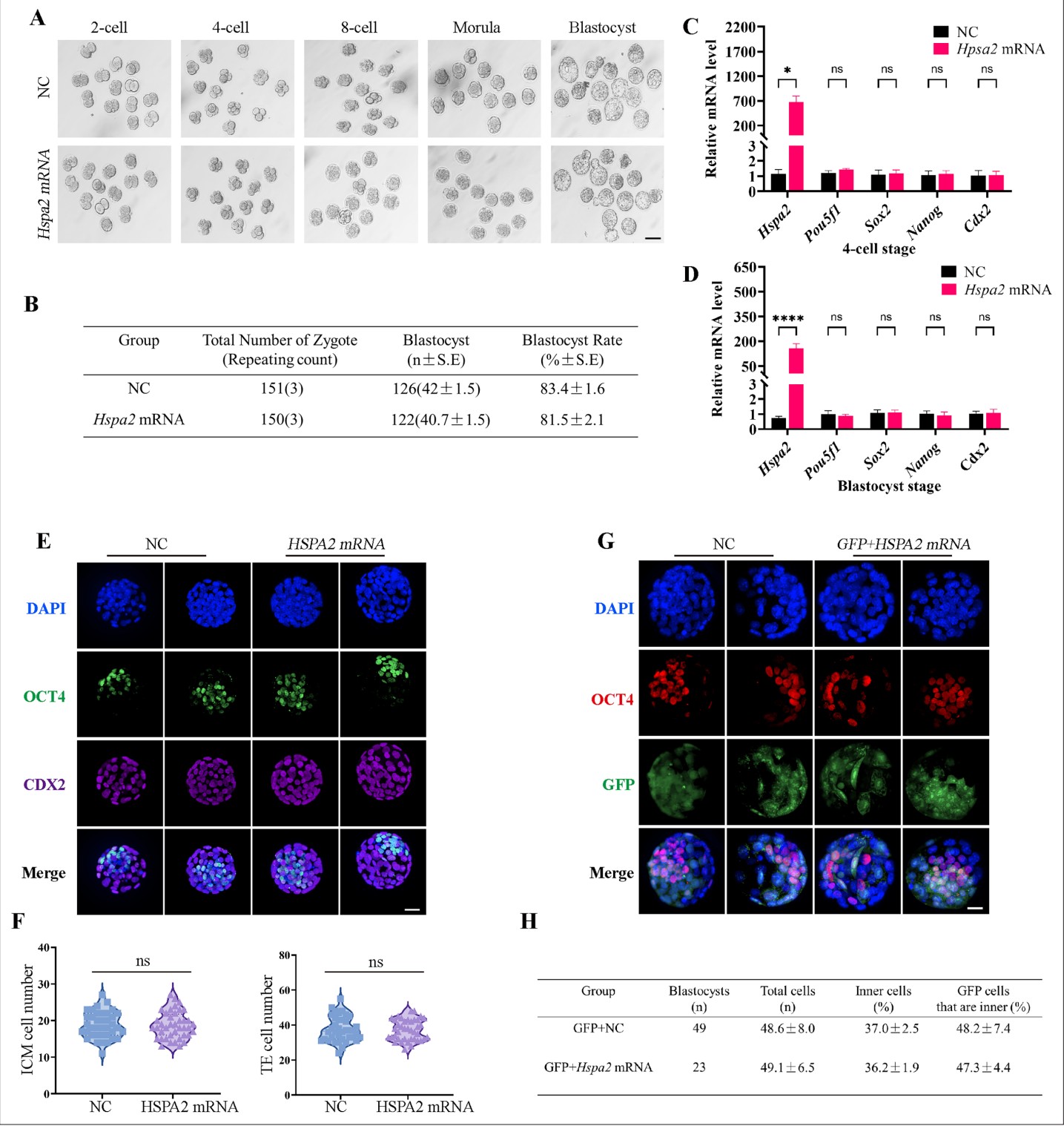

**Figure 4.** Overexpression of *Hspa2* does not induce blastocyst bias towards ICM fate. (**A**) Representative images of preimplantation embryos in the NC and *Hspa2* mRNA groups from 2 cell to blastocyst. Bar, 100 μm. (**B**) The blastocyst rate between NC (n=151) and overexpression (n=150) groups. Three experimental replicates were performed. (**C, D**) qRT-PCR of 4-cell embryo and blastocyst showed that ICM-marker gene (*Pou5f1, Sox2,* and *Nanog*) and TE-marker gene (*Cdx2*) were not significantly different in the overexpression compared with the control group. Three experimental replicates were performed. *p<0.05, ****p<0.001. (**E**) Immunofluorescence of OCT4 (green) and CDX2 (purple) in NC (n=38) and overexpression (n=40) groups. OCT4 was used as an ICM marker. Nuclei were visualized with DAPI staining (blue). Bar, 50 μm. (**F**) The number of ICM and TE. OCT4 was used as an ICM marker and CDX2 was used as a TE marker. (**G**) OCT4 and GFP fluorescent staining of blastocysts. OCT4 was used as an ICM marker. The results show

*Figure 4 continued on next page*

Figure 4 continued

that the percentage of GFP cells that were present in the ICM was similar between the two groups. Three experimental replicates were performed. Bar, 50 μm. (**H**) Analysis of the distribution of progeny of injected blastomere at the blastocyst stage based on fluorescence staining. Total cells (n) are the total number of cells in the blastocyst; inner cells (%) are the percentage of inner cells relative to the total number of cells in the blastocyst; GFP cells that are inner (%) are the percentage of GFP-positive inner cells relative to the total number of GFP-positive cells in the blastocyst. Data derived from three independent experiments and presented as mean ± standard deviation. Three experimental replicates were performed.

The online version of this article includes the following source data and figure supplement(s) for figure 4:

**Figure supplement 1.** Overexpression of *Hspa2* not induces blastocyst bias towards ICM fate.

**Figure supplement 1—source data 1.** PDF file containing original western blot, indicating relevant band.

**Figure supplement 1—source data 2.** Original files for western blot analysis.

blastocyst stage in the *Hspa2*-KD and NC groups by applying untargeted proteomics. Volcano map screening identified a total of 43 annotated DEPs, of which 15 proteins were significantly up-regulated and 28 proteins were down-regulated (*Figure 5—figure supplement 1A*). These DEPs were mainly involved in post-translational modification by using Clusters of Orthologous Groups (COG)/ Karyotic Orthologous Groups (KOG) functional annotation statistics (*Figure 5—figure supplement 1B*). In order to compare the functional similarities and differences between proteins with different differential multiples, we divided the DEPs into four classifications: Q1 (<0.5), Q2 (0.5–0.667), Q3 (1.5–2.0), and Q4 (>2.0) according to differential expression multiples (*Figure 5—figure supplement 1C*). Biological processes analysis showed that DEPs were associated with positive regulation of the Wnt signaling pathway and cellular development (*Figure 5—figure supplement 1D*). In terms of the cellular component, the DEPs were related to the methylome and methyltransferase complex (*Figure 5—figure supplement 1E*). These results indicated that the DEPs were mainly associated with the post-translational modification of proteins and cell fate decision.

Coactivator-associated arginine methyltransferase 1 (CARM1), also known as PRMT4, is the first protein arginine methyltransferase which was proved to be associated with transcriptional activation by methylating and/or regulating histone H3 and non-histone proteins, such as coactivators and transcription factors (*Stallcup et al., 2003*; *Troffer-Charlier et al., 2007*). Recent studies have demonstrated that CARM1 regulates early mouse development and plays a role in the determination of cell fate (*Hupalowska et al., 2018*; *Pawlak et al., 2000*). CARM1 levels, along with its specific histone marks (H3R17me2a and H3R26me2a), were found to vary among the 4-cell blastomeres and that high levels of CARM1 led to increased levels of H3R26me. This varied distribution led to biased subsequent fate of the blastomeres towards the ICM (*Parfitt and Zernicka-Goetz, 2010*; *Torres-Padilla et al., 2007*). Next, we considered whether HSPA2 regulates the decision of the first cell-fate through CARM1. Molecular docking revealed that HSPA2 docked efficiently with CARM1 via several H-bonds and pi-interactions, with strong binding affinity (*Figure 5A*). The Search Tool for the Retrieval of Interacting Genes (STRING) database was used to predict interactions between HSPA2 and CARM1. The results showed that HSPA2, CARM1, and ICM-specific proteins interacted with each other (*Figure 5—figure supplement 1F*). In addition, we found that the *Hspa2*-KD led to reduced CARM1 mRNA as well as protein levels at the 4 cell and blastocyst stages (*Figure 5B–E*). The knockdown of HSPA2 in mESCs also caused a significant reduction in the levels of CARM1 (*Figure 5—figure supplement 1G*).

To further investigate whether HSPA2 could interact with CARM1, we performed co-immunoprecipitation (co-IP) assays in mESCs using HSPA2 and CARM1 as bait. Co-IP analysis showed that HSPA2 and CARM1 proteins interacted effectively (*Figure 5F and G*). Moreover, we investigated whether HSPA2 could regulate H3R26me2 modification, a known event that occurs downstream of CARM1 (*Torres-Padilla et al., 2007*). Immunofluorescence staining confirmed that CARM1-mediated H3R26me2 levels were significantly reduced in the interference group at the 4-cell stage (*Figure 5H*). Collectively, our data demonstrated that HSPA2 involved in the segregation of embryonic cell lineage by forming a complex with CARM1 and through H3R26me2 modification.

## Discussion

During the development of a fertilized egg to blastocyst, the fundamental questions are to explain how the totipotent fertilized egg develops into different cell types and when cellular heterogeneity occurs. Previous studies demonstrated that molecular polarity and differential gene regulation by

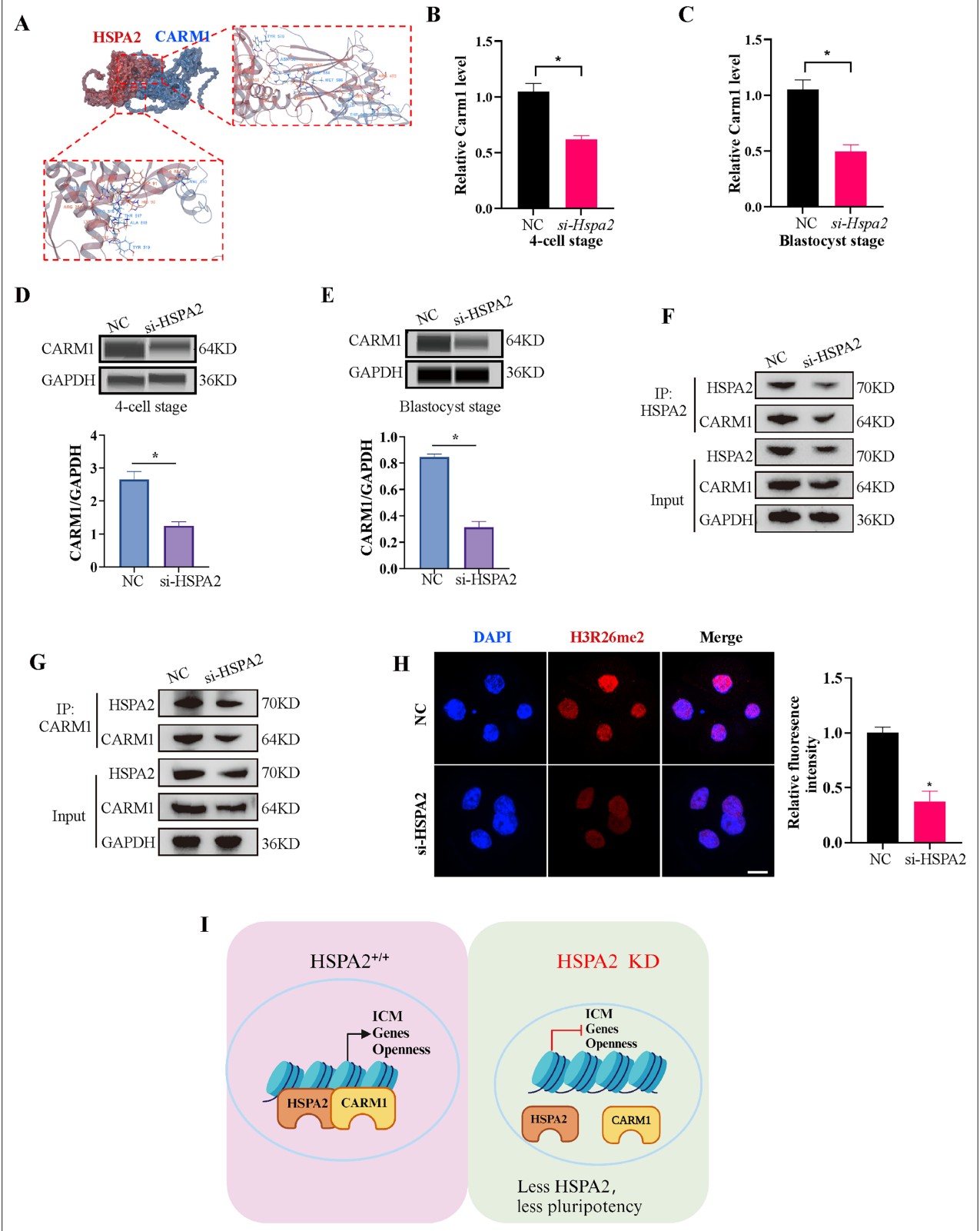

**Figure 5.** HSPA2 and CARM1 form a complex. (**A**) Molecular docking analysis for HSPA2 binding to CARM1 proteins. (**B, C**) qRT-PCR of 4 cell embryo and blastocyst showed that *Carm1* was significantly lower in the interference compared with the control group. Three experimental replicates were performed. *p<0.05. (**D, E**) Automated western blotting analysis of CARM1 expression in the interference and control groups. The band intensity was assessed. Three experimental replicates were performed. *p<0.05 (**F, G**) Co-IP analysis of HSPA2 and CARM1 in mESCs. (**H**) H3R26me2 staining

*Figure 5 continued on next page*

*Figure 5 continued*

of 4-cell embryo shows that HSPA2 knockdown led to a dramatic decrease in H3R26me2 modification. Intensity relative to DAPI signal was used. Three experimental replicates were performed. Bar, 50 µm. *p<0.05. (I) The model of HSPA2 physically binds to CARM1 to activate ICM-specific gene expression. Created in BioRender.

The online version of this article includes the following source data and figure supplement(s) for figure 5:

**Source data 1.** PDF file containing original western blots, indicating relevant band.

**Source data 2.** Original files for western blot analysis.

**Figure supplement 1.** HSPA2 interacts with CARM1 and alters H3R26me2 levels.

**Figure supplement 1—source data 1.** PDF file containing original western blot, indicating relevant band.

**Figure supplement 1—source data 2.** Original files for western blot analysis displayed in.

---

signaling pathways drive the first cell-fate decision in mammals (*Hirate and Sasaki, 2014*; *Johnson and Ziomek, 1981*; *Korotkevich et al., 2017*; *Nishioka et al., 2009*). The tracing of embryonic lineages from the 2-cell stage onwards by live embryo labeling demonstrated that the cellular heterogeneity is initiated early in the embryo, and thus was proposed as cellular heterogeneity hypothesis (*Piotrowska and Zernicka-Goetz, 2001*). H3R26me and its methyltransferase CARM1 exhibit asymmetric expression patterns in the 4-cell stage of mouse embryos and play an important role in regulating the first cell-fate decision. (*Cui et al., 2020*; *Xu et al., 2021*). PRDM14 is also heterogeneously expressed in 4-cell stage mouse embryos and interacts with CARM1 to drive progenies towards pluripotent ICM fate by increasing the expression levels of H3R26me in 4-cell embryos (*Burton et al., 2013*). Despite the progress in our understanding of the mechanisms governing early cell fate decisions, many important questions still remain unanswered. We do not yet fully comprehend when the initial heterogeneities among cells arise and how are they translated into divergent patterns of gene expression (*Chambers et al., 2007*; *Singh et al., 2007*).

Although 2-cell blastomeres are generally considered to be totipotent, the previous reports showing that when 2-cell blastomeres are separated, in the majority of cases, only one of the two blastomeres develops into a mouse (*Casser et al., 2017*). In this study, we found that the expression levels of HSPA2 were symmetrically distributed between the two blastomeres at the early 2-cell stage, but showed heterogeneous distribution at the late 2-cell embryos. Interestingly, *Hspa2* mRNA also heterogeneous distribution at the late 2-cell and 4-cell embryos after microinjection of si-*Hspa2* in zygotes (*Figure 1—figure supplement 1F, G*). This indicates that the heterogeneity of HSPA2 at the late 2-cell embryo may be initiated concomitant with the zygotic genome activation (ZGA) (*Shi et al., 2015*). Alternatively, the asymmetry may be triggered by unequally distributed unknown factors that already existed in the early 2-cell embryos or the heterogeneous spatial distribution of maternal genes in the zygote.

The reduced expression of HSPA2 led to the reduced expression levels of ICM-marker genes at both the mRNA and protein levels. Interfering with HSPA2 expression in one blastomere of the 2-cell embryos reduced the contribution of that blastomere to ICM differentiation. Notably, the number of ICM and total cells in the blastocyst was significantly reduced when compared to the control group (*Figure 3B and E*). We observed that the GO enriched terms were also closely related to negative regulation of cell cycle, G1/S transition of mitotic cell cycle, mitotic cell cycle phase transition and regulation of mitotic cell cycle phase transition (*Figure 2E*). *Hspa2* knockdown via siRNA reduced cell proliferation and led to G1/S phase cell cycle arrest (*Cao et al., 2019*). Thus, we hypothesized that knockdown of HSPA2 results in lower cell number in ICM by interfering with the cell cycle. Additionally, the lower cell number in ICM may also associated with failed cytokinesis and the generation of binucleated or polyploid cells. Interestingly, half of the ICM cells still formed with HSPA2 deficiency (*Figure 3B*), possibly due to the incomplete knockdown achieved or the possibility that redundant pathways exist.

However, the overexpression of HSPA2 did neither adversely affect embryo development nor induce blastomere bias towards an ICM fate. It may be because that once the intracellular concentration of *Hspa2* reaches a threshold level, it was not able to control the target gene expression. Furthermore, we also found that HSPA2 interacted with CARM1, and reduced HSPA2 level, led to the reduced expression of CARM1, and thus reduced the levels of H3R26me2 modification at the 4-cell

stage. Collectively, these results suggest that HSPA2 physically binds to CARM1 to activate ICM-specific gene expression through H3R26me2 modification (*Figure 5I*).

In fact, our RNA-seq results revealed that the DEGs at blastocyst stage with HSPA2 knocking down were also enriched in Wnt signaling pathway and Hippo signaling pathway (*Figure 2E*), which are also reported to be involved in cell proliferation, cell polarity, and cell fate determination during embryonic development (*Nishioka et al., 2009*; *Morris et al., 2012*). This indicates that several alternative mechanisms are compatible with the asymmetric hypothesis, including polarized cell division and differential gene regulation by signaling.

In summary, our findings provide an exciting step forward for the first time that the role of HSPA2 in governing the first cell-fate decision in the mouse embryo lineage allocation. It may be beneficial for achieving a better understanding of embryogenesis patterns and cell differentiation.

## Materials and methods

### Antibodies

In this study, we utilized CDX2 (ab76541), NANOG (ab17338) and H3R26me2 (ab127095) antibodies from Abcam; a CARM1 (12,495 s) antibody from Cell Signaling Technology; SOX2 (11064–1-AP), GAPDH (10494–1-AP), CARM1 (55246–1-AP), HSPA2 (12797–1-AP) and ACTIN (66009–1-Ig) antibodies from Proteintech; and an OCT4 (sc-5279) antibody from Santa Cruz Biotech.

### Mouse embryo collection

Animal experiments were approved by the Ethics Committee of Hospital for Reproductive Medicine affiliated to Shandong University. ICR mice (6–8 weeks-of-age) were purchased from SPF Biotechnology Co.,Ltd. and bred under standard SPF conditions with a 12 hr dark and 12 hr light cycle with food and water provided ad libitum. Superovulation was induced by an intraperitoneal injection of 5 IU of pregnant mare's serum gonadotrophin (PMSG) (110914564, Ningbo Sansheng Biological Technology Co., Ltd) followed 46–48 hr later by 5 IU of human chorionic gonadotrophin (HCG) (110911282, Ningbo Sansheng Biological Technology Co., Ltd.). Oocytes were collected 15 hr after hCG injection. First, the collected oocytes were fertilized in G-IVF (#10,136, Vitrolife) with epididymal sperm from adult males that had been capacitated in G-IVF for 1 hr. All zygotes were cultured in G1 medium (Vitrolife, Sweden) at 37 °C in the presence of 6% $CO_2$ and 5% $O_2$. Early 2-, late 2-, 4-, and 8-cell embryos, and morula- and blastocyst stage embryos were collected after 22–26, 34–38, 48–50, 60–65, 70–75, and 96–100 hr of culture, respectively. Single blastomeres were isolated from 2- and 4-cell embryos and cultured to the morula stage as described previously (*Song et al., 2022*).

### Cell culture and transfection

Mouse ESCs (mESCs) were cultured on plates coated with 0.1% gelatin (Oricell) under feeder-free conditions with 5% CO2 at 37 °C. For cell transfection, the mESCs were transfected with si-*Hspa2* using Lipofectamine 3000 (Invitrogen) according to the manufacturer's instructions. After transfection for 36 hr, cells were collected for further analysis.

**Table 1.** List of mouse primer pairs used for RT-qPCR analysis.

| Gene | Forward primer (5'–3') | Reverse primer (5'–3') |
| --- | --- | --- |
| *Hspa2* | AAGATTTCTTCAACGGCAAGGAG | GGGATCGTGGTGTTTCTCTTGAT |
| *Pou5f1* | GGGTGATGGGTCAGCAG | TCCGCAGAACTCGTATGC |
| *Sox2* | AACGCCTTCATGGTATGG | CTCGGTCTCGGACAAAAG |
| *Nanog* | CTGCTCCGCTCCATAACT | GGCTTTCCCTAGTGGCTT |
| *Cdx2* | GAAGGGGTGGTGGGTTC | AGGTTGGCTCTGGCATTT |
| *Hprt* | GGCTTCCTCCTCAGACCGCTTT | CACTTTTTCCAAATCCTCGGCATAA |

## RT-qPCR analysis

For RT-PCR analysis, we used the Single Cell Sequence Specific Amplification Kit (Vazyme, P621-01) and ChamQ Universal SYBR qPCR Master Mix (Vazyme, Q711-03) in accordance with the manufacturer's instructions. Primers for the candidate genes were designed by online Primer 3.0 software based on the sequence data obtained from the NCBI database. Primer sequences are listed in *Table 1*. Primer sequences purchased from Beijing Tsingke Biotech Co., Ltd. Gene expression was quantified by the comparative CT method. Values are expressed as the mean ± standard error of the mean (SEM). Expression was normalized to Actin. The relative amount of transcript present in each cDNA sample was calculated using the $2^{-\Delta\Delta CT}$ method. All experiments were repeated at least three times, and statistical analysis was performed on individual experimental sets. All measurements were performed in triplicate for each experiment.

## Automated western immunoblotting

Western immunoblots were performed on a PeggySue (ProteinSimple) system using a Size Separation Master Kit with Split Buffer (12–230 kDa) in accordance with the manufacturer's instructions. Capillary-based immunoassays were performed using the Wes-Simple Western method (Proteintech: HSPA2 (12797–1-AP), 1:15 dilution; GAPDH (10494–1-AP),1:35 dilution; SOX2 (11064–1-AP), 1:20 dilution; Abcam: CDX2 (ab76541), 1:10 dilution; Santa Cruz Biotech: OCT4 (sc-5279), 1:15 dilution) and an anti-rabbit or an-mouse detection module (ProteinSimple) (*Schiattarella et al., 2019*). Protein expression was detected by chemiluminescence and quantified by area under the curve (AUC) analysis using the Compass for Simple Western program (ProteinSimple).

## Microinjection and HSPA2 knockdown

Aliquots of 2–5 pl of siRNA at 20 µM concentration were microinjected into the nucleus of zygote or one blastomere of the 2-cell embryos using an Eppendorf (Hamburg, Germany) micromanipulator under a Leica inverted microscope, and then cultured in fresh G1 medium.

To knockdown *Hspa2*, two siRNAs were designed for each gene. The siRNAs of *Hspa2* are GGTG CAACAACTTAGTTTA, GCACTGCAGTGATATTAAA. For microinjection in each KD group, 20 µM of each siRNA was used. siRNAs were ordered from RINOBIO.

## Western blot analysis and co-immunoprecipitation (Co-IP)

Protein concentration was determined with a BCA protein assay (Bio-Rad, Hemel Hempstead, UK). Proteins were then separated by 12% SDS-PAGE and transferred to polyvinylidene difluoride membranes (Millipore, Bedford, MA, USA) at 300mA for 90 min. First, the membranes were incubated in TBST containing 5% non-fat milk for 2 hr; then, the membranes were incubated with primary antibodies (Abcam: CDX2 (ab76541), 1:1000 dilution; NANOG (ab17338) 1:800 dilution; Cell Signaling Technology: CARM1 (12,495 s), 1:1000 dilution; Proteintech: SOX2 (11064–1-AP), 1:800 dilution; GAPDH (10494–1-AP), 1:1500 dilution; HSPA2 (12797–1-AP), 1:1000 dilution; ACTIN (66009–1-Ig), 1:700 dilution; Santa Cruz Biotech: OCT4 (sc-5279), 1:800 dilution) overnight at 4 °C. On the second day, the membranes were washed three times with TBST and then incubated with horseradish peroxidase-conjugated secondary antibody (1:2000 dilution; Proteintech) for 1 hr.

The Pierce Co-IP kit (Thermo Fisher Scientific, USA) was used for co-IP assays. In brief, mESCs were lysed using IP lysis buffer on ice. Next, the lysate was treated with agarose resin at 4 °C for 1 hr followed by incubation with antibodies (Proteintech: HSPA2(12797–1-AP), 1:100 dilution; CARM1(55246–1-AP), 1:60 dilution) at 4 °C overnight. The captured antigen was then eluted and SDS-PAGE was performed.

## RNA FISH and immunofluorescence

Fluorescence in situ hybridization (FISH) was carried out using an RNA-FISH Probe kit (GenePharma, Shanghai), according to the manufacturer's instruction. In brief, embryos were collected and fixed in 4% paraformaldehyde in PBS-DEPC for 15 min at room temperature, permeabilized in 0.1% Buffer A (permeabilizing solution) for 15 min at room temperature, and incubated at 37 °C in 2×Buffer C for 30 min. The embryos were then dehydrated in 70%, 85%, and 100% EtOH and dried at room temperature for 5 min each time. The embryos were denatured in buffer E (hybridization buffer) containing 10 µM probe at 73 °C for 5 min, incubated at 37 °C overnight, and washed in 0.1% Buffer F for 5 min to

remove non-specific binding. After washing in 2×Buffer C and 1×Buffer C (neutral solution) for 5 min, the nuclei were stained with DAPI for 5 min.

To perform immunofluorescence, we removed the zona pellucida with acid Tyrode's solution (Sigma). After that, embryos were fixed in 4% paraformaldehyde in PBS (KGB5001, KeyGEN BioTECH) at room temperature (RT). Embryos were then permeabilized with 1% Triton-100 (T8787, SigmaAldrich) in PBS for 20 min. Embryos were blocked for 30 min at RT on a shaker with a blocking buffer consisting of 1% BSA in PBS containing 0.1% Tween-20 (P9416, Sigma-Aldrich) and 0.01% Triton-100. Next, embryos were incubated with primary antibodies (Abcam: CDX2 (ab76541), 1:150 dilution; H3R26me2 (ab127095), 1:150 dilution; Santa Cruz Biotech: OCT4 (sc-5279), 1:100 dilution) diluted in blocking buffer on a shaker at 4°C overnight. On the second day, embryos were washed three times with TBST and incubated with secondary antibody (1:400 dilution; Sigma) and DAPI (D3571, 1:400 dilution; Life Technologies) for 30 min. First, embryos were rinsed three times for 10 min in washing buffer on a shaker at RT; then, the embryos were mounted on slides in washing buffer and examined under a laser scanning confocal microscope (Dragonfly, Andor Technology, UK). Fluorescence images were acquired at multiple 1 μM intervals in the z-axis with the use of a confocal microscope. Multiple images were processed using ImageJ software for intensity measurement and cell counting (https://imagej.nih.gov/ij/). The fluorescence intensity was quantified by normalization to DAPI using the built-in ImageJ function. The regions of interest (ROI) were selected using a custom Python script. The intensity was measured on normalized sections using the ImageJ measure function. Data were normalized with respect to background levels. To analyze the ICM/TE cell numbers, OCT4 and CDX2 signal at inner layers and outer layers were used to distinguish the ICM and TE cells.

## Molecular docking analysis

For molecular docking analysis, we first generated a PPI network and identified core targets. The proposed protein model was based on the crystal structure downloaded from the AlphaFold Protein Structure Database (https://alphafold.ebi.ac.uk/) and the two- and three-dimensional (2D/3D) structures of potential active compounds were acquired from PubChem (https://pubchem.ncbi.nlm.nih.gov/). Then, molecular docking analysis was performed and visualized in Maestro11.9; the resultant docking score was saved and then compared.

## Proteomics and bioinformatic analysis

For proteomics and bioinformatic analysis, 4 embryos per sample were collected. Four replicates were assessed for NC group and two replicates were assessed for KD group. Protein samples were sonicated three times on ice using a high intensity ultrasonic processor (Scientz) in lysis buffer (8 M urea, 1% protease inhibitor cocktail). Inhibitors were also added to lysis buffer for PTM experiments. The remaining debris was removed by centrifugation at 12,000 × $g$ at 4 °C for 10 min. The protein samples were then diluted by adding 200 mM of TEAB to urea concentrations <2 M. Finally, trypsin was added at a 1:50 trypsin-to-protein mass ratio for the first digestion overnight and a 1:100 trypsin-to-protein mass ratio for a second 4 h-digestion. Finally, the peptides were desalted with a Strata X SPE column. The eluted peptides were then cleaned with C18 ZipTips (Millipore) in accordance with the manufacturer's instructions, followed by analysis with LC–MS/MS. The resulting MS/MS data were processed by MaxQuant with an integrated Andromeda search engine (version 1.4.1.2). False discovery rate thresholds for proteins, peptides and modification sites were specified at 1%.

COG, full name is Clusters of Orthologous Groups of proteins. The proteins that make up each COG are assumed to be derived from an ancestral protein. Orthologs are proteins that are derived from different species, evolved from vertical lineages (species formation), and typically retain the same function as the original protein. COG means 'cluster of homologous proteins' in Chinese. COGs are divided into two categories, one for prokaryotes and the other for eukaryotes. Those for prokaryotes are generally referred to as COG databases; those for eukaryotes are generally referred to as KOG databases.

## RNA-seq analysis

For RNA sequencing, 3 embryos per sample were collected. Three replicates were assessed for NC group and three replicates were assessed for KD group. After being denuded of the zona pellucida using EmbryoMax Acidic Tyrode's Solution (MR-004-D, Sigma-Aldrich), RNA-seq libraries were generated

following the Smart-seq2 protocol as described previously (*Picelli et al., 2014*). Sequencing libraries were generated using TruePepDNA Library Prep Kit V2 for Illumina (TD502, Vazyme) according to manufacturer's recommendations. The libraries were performed on an Illumina Hiseq X-ten platform with 150 bp paired-end.

Filtering out low quality and joint sequences, the clean reads of each sample obtained were mapped to the reference genome downloaded from the genome website (https://ftp.ensembl.org/pub/release-105/fasta/mus_musculus/dna/Mus_musculus.GRCm39.dna.primary_assembly.fa.gz) using Hisat2. The expression values for each gene were calculated as Fragments Per Kilobase of exon model per Million mapped fragments (TPM +1). DEG analysis was conducted with edgeR package. In this project, the genes were identified as differentially expressed genes (DEGs) if |log2FC|>1 and p<0.05 (FC: fold change). Gene Ontology (GO) and Kyoto Encyclopedia of Genes and Genomes (KEGG) enrichment analyses were s carried out for all the DEGs, utilizing the 'cluster Profiler' package.

## Statistical analysis

Statistical analyzes were performed by GraphPad Prism software (version 9.3, GraphPad Prism Software Inc, San Diego, CA) and the Statistics Package for Social Science (SPSS 11.5; SPSS Inc, Chicago, IL, USA). Statistical significance (*Figure 1E-J*, *Figure 2I-M*, *Figure 3B, E*, *Figure 4B-D, F*, *Figure 5B-E, H*, *Figure 1–figure supplement 1C-G*, *Figure 5–figure supplement 1G*) were calculated with two tailed Student's t test. Levels of significance (*Figure 1C*) were calculated with t test. At least three biological replicates were performed for each experiment and results are represented as mean ± standard error of mean (SEM). p<0.05 was considered statistically significant.

## Acknowledgements

This work was supported by National Natural Science Foundation of China (32170817); Fundamental Research Funds for the Central Universities (2023QNTD004, 2022JC006); Taishan Scholars Program of Shandong Province (tsqn202211373); Innovative research team of high-level local universities in Shanghai (SHSMU-ZLCX20210201); Shandong Provincial Natural Science Foundation (2022HWYQ-034, ZR2021QC106).

## Additional information

### Funding

| Funder | Grant reference number | Author |
|---|---|---|
| National Natural Science Foundation of China | 32170817 | Keliang Wu |
| Fundamental Research Funds for the Central Universities | 2023QNTD004 | Keliang Wu |
| Fundamental Research Funds for the Central Universities | 2022JC006 | Keliang Wu |
| Taishan Scholars Program of Shandong Province | tsqn202211373 | Boyang Liu |
| Innovative research team of high-level local universities in Shanghai | SHSMU-ZLCX20210201 | Keliang Wu |
| Shandong Provincial Natural Science Foundation | 2022HWYQ-034 | Boyang Liu |
| Shandong Provincial Natural Science Foundation | ZR2021QC106 | Boyang Liu |

| Funder | Grant reference number | Author |
|--------|------------------------|--------|

The funders had no role in study design, data collection and interpretation, or the decision to submit the work for publication.

## Author contributions
Jiayin Gao, Conceptualization, Data curation, Writing – original draft, Writing – review and editing; Jiawei Wang, Formal analysis; Shiyu Liu, Conceptualization, Methodology; Jinzhu Song, Chuanxin Zhang, Methodology; Boyang Liu, Conceptualization, Funding acquisition, Methodology; Keliang Wu, Conceptualization, Funding acquisition, Methodology, Project administration

## Author ORCIDs
Jiayin Gao ⓘ https://orcid.org/0009-0001-4940-3914
Jiawei Wang ⓘ https://orcid.org/0000-0003-3231-0305
Shiyu Liu ⓘ https://orcid.org/0009-0006-1879-6635
Boyang Liu ⓘ https://orcid.org/0000-0002-1285-4842

## Ethics
The current study was carried out with the approval of the Ethics Committee of Hospital for Reproductive Medicine affiliated to Shandong University (2021-102), and all experiments performed were abided by relevant regulations and guidelines of the committees.

Reviewer #1 (Public review): https://doi.org/10.7554/eLife.100730.3.sa1
Reviewer #2 (Public review): https://doi.org/10.7554/eLife.100730.3.sa2
Author response https://doi.org/10.7554/eLife.100730.3.sa3

# Additional files

## Supplementary files
MDAR checklist

## Data availability
Sequencing data have been deposited in GEO under accession codes GSE266025. Proteomic data are available via ProteomeX change with identifier PXD052193.

The following datasets were generated:

| Author(s) | Year | Dataset title | Dataset URL | Database and Identifier |
|-----------|------|---------------|-------------|-------------------------|
| Gao J | 2024 | The asymmetric expression of HSPA2 in blastomeres governs the first embryonic cell-fate decision | https://www.ncbi.nlm.nih.gov/geo/query/acc.cgi?acc=GSE266025 | NCBI Gene Expression Omnibus, GSE266025 |
| Gao J | 2025 | differentially expressed proteins (DEPs) at the blastocyst stage in the Hspa2-KD and NC groups by applying proteomics technology | https://proteomecentral.proteomexchange.org/cgi/GetDataset?ID=PXD052193 | ProteomeXchange, PXD052193 |

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
