## [Editor Report · eLife Assessment]

This **useful** study by Gao et al identifies Hspa2 as a heterogeneous transcript in the early embryo and proposes a plausible mechanism showing interactions with Carm1. The authors propose that variability in HSPA2 levels among blastomeres at the 4-cell stage skews their relative contribution to the embryonic lineage. Given only 4 other heterogeneous transcripts/non-coding RNA have been proposed to act similarly at or before the 4-cell stage, this would be a key addition to our understanding of how the first cell fate decision is made. While this is a **solid** study, further data are needed to fully support the conclusions.

---

## [Referee Report · Reviewer #1 (Public review)]

Summary:

The authors investigate the role of HSPA2 during mouse preimplantation development. Knocking down HSPA2 in zygotes, the authors describe lower chances of developing into blastocysts, which show a reduced number of inner cell mass cells. They find that HSPA2 mRNA and protein levels show some heterogeneity among blastomeres at the 4-cell stage and propose that HSPA2 could contribute to skewing their relative contribution to embryonic lineages. To test this, the authors try to reduce HSPA2 expression in one of the 2-cell stage blastomere and propose that it biases their contribution to towards extra-embryonic lineages. To explain this, the authors propose that HSPA2 would interact with CARM1, which controls chromatin accessibility around genes regulating differentiation into embryonic lineage.

Strengths:

(1) The study offers simple and straightforward experiments with large sample sizes.

(2) Unlike most studies in the field, this research often relies on both mRNA and protein levels to analyse gene expression and differentiation.

Weaknesses:

(1) Image and statistical analyses are not well described.

(2) The functionality of the overexpression construct is not fully validated.

(3) Tracking of KD cells in embryos injected at the 2-cell stage with GFP is unclear.

(4) A key rationale of the study relies on measuring small differences in the levels of mRNA and proteins using semi-quantitative methods to compare blastomeres. As such, it is not possible to know whether those subtle differences are biologically meaningful. For example, the lowest HSPA2 level of the embryo with the highest level is much higher than the top cell from the embryo with the lowest level. What does this level mean then? Does this mean that some blastomeres grafted from strong embryos would systematically outcompete all other blastomeres from weaker embryos? That would be very surprising. I think the authors should be more careful and consider the lack of quantitative power of their approach before reaching firm conclusions. Although to be fair, the authors only follow a long trend of studies with the same intrinsic flaw of this approach.

(5) Some of the analyses on immunostaining do not take into account that this technique only allows for semi-quantitative measurements and comparisons.

a) Some of the microscopy images are shown with an incorrect look-up table.

b) Some of the schematics are incorrect and misleading.

---

## [Referee Report · Reviewer #2 (Public review)]

Summary:

In this study, Gao et al. use RNA-seq to identify Hspa2 as one of the earliest transcripts heterogeneously distributed between blastomeres. Functional studies are performed using siRNA knockdown showing Hspa2 may bias cells toward the ICM lineage via interaction with the known methyltransferase CARM1.

Strengths:

This study tackles an important question regarding the origins of the first cell fate decision in the preimplantation embryo. It provides novelty in its identification of Hspa2 as a heterogeneous transcript in the early embryo and proposes a plausible mechanism showing interactions with Carm1. Multiple approaches are used to validate their functional studies (FISH, WB, development rates, proteomics). Given only 4 other transcripts/RNA have been identified at or before the 4-cell stage (LincGET, CARM1, PRDM14, HMGA1), this would be an important addition to our understanding of how TE vs ICM fate is established.

Weaknesses:

The RNA-seq results leading the authors to focus on Hspa2 are not included in the manuscript. This dataset would serve as an important resource but is neither included nor discussed. Nor is it mentioned whether Hspa2 was identified in prior RNA-seq embryos studies (for example Deng Science 2014).

Furthermore, the authors show that Hspa2 knockdown at the 1-cell stage lowers total Carm1 levels at the 4-cell stage. However, it is unclear how total abundance within the embryo alters lineage specification within blastomeres. The authors go on to propose a plausible mechanism involving Hspa2 and Carm1 interaction, but do not discuss how expression levels may be involved.

---

## [Author Response]

The following is the authors’ response to the original reviews.

**Public Reviews:**

**Reviewer #1 (Public review):**
Summary:The authors investigate the role of HSPA2 during mouse preimplantation development. Knocking down HSPA2 in zygotes, the authors describe lower chances of developing into blastocysts, which show a reduced number of inner cell mass cells. They find that HSPA2 mRNA and protein levels show some heterogeneity among blastomeres at the 4-cell stage and propose that HSPA2 could contribute to skewing their relative contribution to embryonic lineages. To test this, the authors try to reduce HSPA2 expression in one of the 2-cell stage blastomere and propose that it biases their contribution to towards extra-embryonic lineages. To explain this, the authors propose that HSPA2 would interact with CARM1, which controls chromatin accessibility around genes regulating differentiation into embryonic lineage.Strengths:(1) The study offers simple and straightforward experiments with large sample sizes.

Thanks for your kind recognition.

(2) Unlike most studies in the field, this research often relies on both mRNA and protein levels to analyses gene expression and differentiation.

Thanks for your kind recognition.

Weaknesses:(1) Image and statistical analyses are not well described.

Thanks for your advisable comment. We redescribe the image and statistical analyses in our revised version (line 255-257).

(2) The functionality of the overexpression construct is not validated.

Thanks for your kind suggestion. We validate the functionality of the overexpression construct in our revised version (Figure S3).

(3) Tracking of KD cells in embryos injected at the 2-cell stage with GFP is unclear.

Thanks for your kind suggestion. We randomly co-injected green fluorescent protein (Gfp) mRNA as a linage tracer with either Hspa2-siRNA or NC-FAM into one of the 2 -cell, and then monitored embryo development to the blastocyst stage (line 342-344).

(4) A key rationale of the study relies on measuring small differences in the levels of mRNA and proteins using semi-quantitative methods to compare blastomeres. As such, it is not possible to know whether those subtle differences are biologically meaningful. For example, the lowest HSPA2 level of the embryo with the highest level is much higher than the top cell from the embryo with the lowest level. What does this level mean then? Does this mean that some blastomeres grafted from strong embryos would systematically outcompete all other blastomeres from weaker embryos? That would be very surprising. I think the authors should be more careful and consider the lack of quantitative power of their approach before reaching firm conclusions. Although to be fair, the authors only follow a long trend of studies with the same intrinsic flaw of this approach.

Thanks for your advisable comment. Indeed, despite the approach drew on previous research (Zhou Cell 2018), we were clearly aware that this approach can only reflect relative comparisons. This means that the relative difference among the blastomeres from the same embryo were detected and compared. We did not compare the absolute levels of mRNA between different embryos. We also offered simple and straightforward experiments with large sample sizes to confirm this conclusion.

(5) Some of the analyses on immunostaining do not take into account that this technique only allows for semi-quantitative measurements and comparisons.a) Some of the microscopy images are shown with an incorrect look-up table.b) Some of the schematics are incorrect and misleading.

Thanks for your advisable comment. We revised microscopy images and schematics in our revised version.

**Reviewer #2 (Public review):**
Summary:In this study, Gao et al. use RNA-seq to identify Hspa2 as one of the earliest transcripts heterogeneously distributed between blastomeres. Functional studies are performed using siRNA knockdown showing Hspa2 may bias cells toward the ICM lineage via interaction with the known methyltransferase CARM1.Strengths:This study tackles an important question regarding the origins of the first cell fate decision in the preimplantation embryo. It provides novelty in its identification of Hspa2 as a heterogeneous transcript in the early embryo and proposes a plausible mechanism showing interactions with Carm1. Multiple approaches are used to validate their functional studies (FISH, WB, development rates, proteomics). Given only 4 other transcripts/RNA have been identified at or before the 4-cell stage (LincGET, CARM1, PRDM14, HMGA1), this would be an important addition to our understanding of how TE vs ICM fate is established.

Thanks for your kind recognition.

The RNA-seq results leading the authors to focus on Hspa2 are not included in the manuscript. This dataset would serve as an important resource but is neither included nor discussed. Nor is it mentioned whether Hspa2 was identified in prior RNA-seq embryos studies (for example Deng Science 2014).

Thanks for your advisable comment. To identify genes that show a significantly high variability across blastomeres in the same embryo, we regressed out the embryo effect by established a new method, which will be published and uploaded to the database in the future. Thus, the RNA-seq results leading the we focus on Hspa2 are not included in the manuscript.

In addition, the functional studies are centered on Hspa2 knockdown at the zygote (1-cell) stage, which would largely target maternal transcript. Given the proposed mechanism relies on Hspa2 heterogeneity post-ZGA (late 2-cell stage), the knockdown studies don't necessarily test this and thus don't provide direct support to the authors' conclusions. The relevance of the study would be improved if the authors could show that zygotic knockdown leads to symmetric Hspa2 levels at the late 2-cell and/or 4-cell stage. It may be possible that zygotic knockdown leads to lower global Hspa2 levels, but that asymmetry is still generated at the 4-cell stage.

Thanks for your advisable comment. We showed that the *Hspa2* levels at the late 2-cell and 4cell stage after zygotic knockdown in our revised version (Figure S1 G-H, line 450-452).

Furthermore, the authors show that Hspa2 knockdown at the 1-cell stage lowers total Carm1 levels at the 4-cell stage. However, it is unclear how total abundance within the embryo alters lineage specification within blastomeres. The authors go on to propose a plausible mechanism involving Hspa2 and Carm1 interaction, but do not discuss how expression levels may be involved.

Thanks for your advisable comment. Previous research suggests that heterogeneous activity of the methyltransferase CARM1 results in differential methylation of histone H3R26 to modulate establishment of lineage specification (Zernicka-Goetz Cell 2018). Thus, we didn't discuss the total abundance within the embryo alters lineage specification.

**Recommendations for the authors:**

**Reviewer #1 (Recommendations for the authors):**
(1) Major issue with analyses:Image analysis needs to be much better explained than simply saying that ImageJ was used. Where are cells measured (at their equatorial plane? What is the size of the ROI?)? Ideally, the ROI and/or raw measurements should be provided.

Thanks for your advisable comment. We redescribe the Image analysis in our revised version (line 187-194).

What are the objective criteria determining whether a cell is counted as GFP positive, CDX2 positive, or OCT4 positive? This is very unclear and key to the interpretation of many experiments.

Thanks for your advisable comment. We think that the cell containing fluorescence signals above background noise were counted positive.

Statistical analyses mention ANOVA in the methods but the student's t-test in the figure legend. Which is which? Most data are heavily normalized, which would unlikely fit the description for Student's t-test analyses.

Thanks for your advisable comment. We redescribe the statistical analyses in our materials and methods (line 253-260).

Figure 5H describes a relative fluorescence intensity with control at 1. The legend describes a normalization to "DNA" (I guess the authors meant DAPI), which is unlikely to give 1. This suggests that additional normalization was done and is not described. Is that the case? Also, since the authors propose that HSPA2 would control Histone modification and chromatin packing, I do not think that using DAPI is an appropriate way of normalizing the fluorescence signal.

Thanks for your advisable comment. We replaced DNA with DAPI in our revised version. Based on previous studies, we adopted DAPI as a normalized fluorescence signal (Zhou Cell 2018, Zernicka-Goetz Cell 2018).

Figure 1E shows data normalized to the lowest level while Figure 1H is normalized to the highest level. A consistent representation would be welcome.

Thanks for your advisable comment. We revised the Figure 1H in our revised version.

Is Figure 1C showing a t-test between correlations?

Yes, Figure 1C shows the t-test between correlation.

(2) Major issue with the interpretation of semi-quantitative methods and measurements:qPCR, WB, immunostaining are all semi-quantitative methods that require some kind of normalization due to non-linear bias in the way the molecules are picked up. Such normalization makes it difficult to know whether a detectable difference is meaningful biologically speaking i.e. if a difference of 1 CT between blastomeres can be detected after qPCR, is it meaningful? If that were the case, then embryos with lower CT than others (Figure 1D) would not be able to develop into blastocyst, like siRNA injected embryos, or grafting a blastomere with a high CT onto an embryo with low CT would lead to the systematic differentiation of these strong blastomeres into ICM.

Thanks for your advisable comment. The CT values represent the relative mRNA levels of *Hspa2* between blastomeres, and the higher CT value represents the lower expression of *Hspa2* at mRNA level. Figure 1D shows the *Hspa2* mRNA levels between blastomeres. The blastomere with lowlevel expression of the *Hspa2* mRNA is not bias an ICM fates.

The same goes for fluorescence analyses (Figure 1F). Can the authors also provide the measurements for DAPI as they did for HSPA2? I am sure that with enough measurements, DAPI is variable enough to give a statistical difference among blastomeres with questionable biological meaning.I think the reasoning used here (unfortunately following the reasoning that has been used in a series of studies by other groups) of ranking blastomeres after semi-quantitative measurement is fundamentally flawed.

Thanks for your advisable comment. The DAPI was determined by the maximal area using a custom Python script. Based on previous studies, we adopted DAPI as a normalized fluorescence signal (Zhou Cell 2018). This approach is to normalize embryo-to-embryo variance from the technical reason.

(3) Major issue with overexpression experiment:While the siRNA experiment is partially validated by qPCR and WB measurements of HSPA2 after KD, the overexpression experiment is not. Do the authors have any evidence that the construct they use is produced into protein and functional? Can the authors check by WB? Can the authors rescue the siRNA with their overexpression?

Thanks for your advisable comment. We verified the overexpression experiment by WB in in our revised version (Figure S3, line 360-361). Considering that siRNA degrades mRNA and prevents the mRNA translation process, we did not co-inject the siRNA with their overexpression.

The lack of effect of HSPA2 overexpression on blastocyst formation is difficult to reconcile with the interpretation from the authors that levels of HSPA2 bias lineages.

Have the authors tried lower concentrations? Have the authors tried FISH on their half-injected 2cell embryos? Of course, if the antibody against HSPA2 would work with immunostaining, that would be ideal.

Thanks for your advisable comment. We chose the concentrations for our study based on previous research (Zernicka-Goetz Cell 2016). To verified Hspa2 was successfully inject into one blastomere at the 2-cell stage, we observed green fluorescence after co-injected GFP mRNA with either siRNA or NC-FAM into one blastomere of the two-cell embryos. Thus, we didn't try FISH on half-injected 2-cell embryos. We tried to perform immunostaining experiments with various HSPA2 antibodies (Proteintech: 12797-1-AP, Abcam: ab108416) and no good results were achieved.

(4) Major issue with tracking of injected cells:It is unclear what counts as a GFP-positive cell. In Figure 3D, most cells appear to have the same level of GFP.

Thanks for your advisable comment. The cell containing green fluorescence signals above background noise were counted GFP-positive in Figure 3D. Most cells seem to have the same level of GFP because they are daughter cells of the blastomeres injected with GFP.

In the images of GFP-expressing cells used to track the control of KD cells shown in Figure 3A, it seems that the control embryos have mostly GFP cells in the ICM. Is that the case, or just a bad example?

Thanks for your advisable comment. The green fluorescent signals in Figure 3A represented OCT4 protein, an ICM marker.

Can the authors do FISH against HSPA2 and visualize their GFP cells to validate the heterogeneous expression in situ?

Thanks for your advisable comment. We have verified the heterogeneous expression of HSPA2 in Figure1.

(5) Issue with fluorescent images:Many images are shown with inappropriate look-up tables with saturated DAPI, OCT4, CDX2, and FISH. This raises the doubt that analyses were made on saturated images, which would be incorrect.

The LUT of Figure 5H should be adjusted similarly between the control and siRNA.

Thanks for your advisable comment. We revised some images which showed inappropriate lookup tables in our revised version. The LUT of Figure 5H had been adjusted between the control and siRNA.

(6) Issue with schematics:Schematics of blastomere isolation grown into blastocyst-like structures are misleading since the final blastocyst-like structure should not have a zona pellucida and should have fewer cells than regular blastocysts.

Thanks for your advisable comment. We revised schematics of blastomere grown into morula in our revised version (Figure 1A and Figure S1A).

The summary schematics in the final figure should not state HSPA2 -/- since experiments in the study did not use KO but KD.

Thanks for your advisable comment. We revised the summary schematics in our revised version.

The blastocysts are the same sizes as the cleavage stage or morula embryos which implies that cells lose volume to the lumen, which is not the case.

Thanks for your advisable comment. We revised the schematics in our revised version.

(7) Issue with data presentation:In the tables within the figures, the number of decimals given should be the same for the mean and SE (one decimal should be more than enough).

Thanks for your advisable comment. We revised the figure 2H in our revised version.

The comparison of cell number and distribution within embryos (e.g. Figure 2B) would be best represented by a correlation analysis of TE vs ICM cells.

Thanks for your advisable comment. We add the figure of a correlation analysis of TE vs ICM cells in our revised version (Figure 3B).

The docking simulations are described in the main text as "experiments".

Thanks for your advisable comment. We redescribed the docking simulations in our revised version.

(8) Issue with data interpretation:The reduced number of ICM cells is interpreted as a slowed-down cell cycle. This could also be explained by failed cytokinesis and the generation of binucleated or polyploid cells. Have the authors checked for that? For example, by looking at their DAPI staining.

Thanks for your advisable comment. Our RNA-seq results revealed that the differentially expressed genes (DEGs) at blastocyst stage with HSPA2 knocking down are closely related to negative regulation of cell cycle, G1/S transition of mitotic cell cycle, mitotic cell cycle phase transition and regulation of mitotic cell cycle phase transition. Additionally, the previous study demonstrated that knockdown of HSPA2 reduced cell proliferation and led to G1/S phase cell cycle arrest (Hu Ann Transl Med 2019). Additionally, the lower cell number in ICM may also associated with failed cytokinesis and the generation of binucleated or polyploid cells. Thus, we guessed that HSPA2 has a role in ICM lineage establishment, although half of the ICM cells were able to survive with HSPA2 deficiency (line 463-472).

It is unclear to me why reduced ICM should lead to fewer blastocysts. Blastocysts should be able to form as long as their TE is fine. In Figure 2G, embryos seem to be cultured in close proximity, which is fine if they are healthy but not if some of the embryos start dying and releasing toxic compounds (e.g. ROS). Have the authors tried removing the dying KD embryos to see if the development of the remaining embryos would improve?

Thanks for your advisable comment. We think HSPA2 may affect blastocyst development by affecting other signaling pathways. And, the GO enriched terms was closely related to blastocyst development (Figure 2E). There was no significant difference in morula formation rate between *Hspa2*-KD group and NC group, thus the assumption that the toxic compounds released by some of the embryos that lead to downregulation of blastocyst rate may not be correct. Indeed, the rate of blastocyst formation in *Hspa2*-KD embryos was reduced significantly lower when few embryos was cultured separately. In addition, we discussed the possibility that the lower cell number in ICM may also associated with failed cytokinesis and the generation of binucleated or polyploid cells.

**Author response image 2. sa3fig2:** 

**Reviewer #2 (Recommendations for the authors):**
One of the significant findings in the paper is the discovery portion where Hspa2 is identified as a heterogeneous transcript. To improve the logic and impact of the manuscript, it may benefit from reorganizing some of the figures and text. For example:(1) The paragraph in the introduction (Lines 56-68) should be moved to the discussion as the Hspa2 reveal should be in section 3.1, not prior to the RNA-seq results presented in Figure 1.

Thanks for your advisable comment. We think it is more logical that HSPA2 needs to be introduced in the introduction.

(2) Add text at the beginning of Section 3.1 to describe the rationale and results for the RNAseq. It would help the readers if the authors clearly stated why they chose the 4-cell stage.

Thanks for your advisable comment. We explain why we chose the 4-cell stage in our revised version (line 272-273).

(3) As this is the first time Hspa2 is identified, consider moving Figure S1C to the main figure to show expression throughout development.

Thanks for your advisable comment. We moved Figure S1C to the main figure in our revised version (line 286-291).

(4) Figure 1C: the correlation between Hspa2 and ICM markers would be strengthened if additional transcripts were used (Oct4, Sox2, Sox21). The graph in 1C would also be more informative if represented as a scatter plot with correlation coefficients (Nanog log2TPM vs Hspa2 log2TPM), rather than bar graphs.

Thanks for your advisable comment. We chose Nanog as the correlation between Hspa2 and Nanog, a ICM markers, was showing the strongest correlation in result. And, the figure 1C shows the stronger positive correlation between *Nanog* and *Hspa2* in gene expression than random gene pairs (n=100, n means the number of random gene pairs). Thus, the figure 1C with bar graphs is easier to understand.

(5) Figure 1D: how were individual blastomeres grouped into B1-4? Individually run and then pooled based on relative expression?

Thanks for your advisable comment. Blastomeres are named B1 to B4 according to increasing Hspa2 concentration in figure 1E.

(6) Figures 1F, 1I, 5H: the DAPI channel appears to be saturated, but is used to normalize fluorescence intensity and may incorrectly account for light scattering within the embryo. Please clarify by adding more details regarding image analysis. Were partial stacks through the nucleus used for analysis, or max projections? Graph axes should be "relative fluorescence intensity."

Thanks for your advisable comment. We added the details of fluorescence images analysis. The graph axes had revised in our revised version.

(7) Line 278: the results in Figure S1C would benefit from more text regarding expression patterns throughout development. The maternal transcript appears to have a sharp downregulation by the early 2-cell stage, and is then upregulated coinciding with ZGA.

Thanks for your advisable comment. We added more describe of the Figure in main text (LINE 285-290).

(8) For the analyses in Figure 2 I-J and 2K-L, were arrested embryos excluded from analysis? This is an important detail as including arrested embryos would significantly bias the RNA-seq results.

Thanks for your advisable comment. The arrested embryos were excluded in Figure 2 I-J and 2K-L.

(9) Figures 2G-H would be aided by converting the table in 2H to a bar graph and adding development rates for all stages (2-, 4-, 8-, morula, and blast). This would also show when an arrest occurs.

Thanks for your advisable comment. We converted the table in 2H to a bar graph.

(10) Blast rates are represented with too many significant digits (Figures 2H, 4B). They should only be reported to the closest ones given the unit of measure (number of blasts divided by number of zygotes). For instance, a blast rate of 81.63 {plus minus} 2.000 reflects excessive precision that is not measured in the data, it should rather read 82 {plus minus} 2%. This is also true for % cells (Figures 3E, 4H).

Thanks for your advisable comment. Values were rounded down to the one decimal place (rounded down).

(11) The clarity and impact of Figure 3A and 3D would benefit from 2D slices through the ICM.

Thanks for your advisable comment. In order to get more comprehensive understanding of the 3D structure of blastocyst of Figure 3A and 3D, we did not choose 2D slices.

(12) To improve clarity and logic, separate the 1-cell and 2-cell knockdown experiments in the text and figures:a) 1-cell knockdown with RNA-seq results (Fig 2A-F).b) 1-cell knockdown showing less ICM/pluripotency markers in (combine Figures 2G-M and Figures 3A-B; "new Fig 3").c) 2-cell knockdown tracing lineage (Figures 2D-E; "new Fig 4").The new Figures 3 and 4 should mirror one another (i.e. for each knockdown experiment, development rates and cell counts should be included). For the 2-cell knockdown (Figures 2 D-E), what were the developmental rates (8-cell, morula, blast)?

Thanks for your advisable comment. However, in order to the overall logical of the article, we do not separate the 1-cell and 2-cell knockdown experiments in the text and figures. And, we added the developmental rates (8-cell, morula, blast) of 2-cell knockdown group in our revised version (Figure S2).

For the overexpression experiment (Figure 4), why were injections performed at the zygote stage versus the 2-cell stage? Given the significant downregulation of maternal transcript demonstrated in Figure S1C, it seems plausible that the injected RNA was also downregulated.

Thanks for your advisable comment. For the overexpression experiment, we first chose to inject *Hspa2* mRNA at the zygote stage and found that the overexpression of Hspa2 does not induce blastomere cells to bias an ICM fate. The qRT-PCR results indicated that the expression level of *Hspa2* in overexpression group was significantly increased compared with normal group at 4cell and blastocyst stage (Figure 4C, 4D). In addition, there is no guarantee that an equal amount of *Hspa2* mRNA be injected into each blastomere in 2-cell stage. Thus, we did not microinject *Hspa2* mRNA into the 2-cell stage.

The 3.5 subheading overstates the results as the Hspa2-Carm1 interaction is not linked to lineage segregation. For example, a more specific subtitle might be, "Hspa2 interacts with Carm1 and alters H3R26me2 levels."

Thanks for your advisable comment. We revised the subtitle in our revised version (line 376).

Figures 5B-C and 5D-E. The qRT-PCR and WB analysis of knockdown blasts shows a correlation between Hspa2 downregulation and Carm1 downregulation. However, if the proposed mechanism is Hspa2 binding to Carm1 to mediate downstream methylation, why would it be expected to alter transcript levels at the 4-cell or blast stage? Please add further details and discussion in the results and discussion sections.

Thanks for your advisable comment. The reason we chose to work at the 4-cell stage is because previous studies on CARM1 have focused on the 4-cell stage (Zernicka-Goetz Cell 2018,2016).

In the discussion, the statement in Lines 430-431 is an overinterpretation: "the heterogeneity of HSPA2... acts as an upstream factor to drive [the] first cell-fate decision." The knockdown experiments don't alter heterogeneity per se, but total abundance. Furthermore, the results do not show that heterogeneity drives heterogeneity in H3R26me2 patterns, for example.

Thanks for your advisable comment. We redescribe the relevant statement in the discussion.

More needs to be said regarding the ICM cells that persisted in the 1-cell KD experiment (Fig 3B). Lines 449-450 point out this result, but do not propose any plausible explanations. For instance, ICM cells may still form due to the incomplete knockdown achieved or the possibility that redundant pathways exist.

Thanks for your advisable comment. We redescribe the relevant statement in our revised version (line 468-473).

The 5th paragraph of the discussion seems incomplete. The authors point out a possible link between Hspa2 and Hippo and Wnt signaling pathways, but need to expand their discussion on how this may act as an additional mechanism incorporating Hspa2 with lineage segregation.

Thanks for your advisable comment. We redescribe the 5th paragraph of the discussion (line 483-494).

Statistics: all comparisons with greater than 2 groups should be performed with a one-way ANOVA and multiple comparisons, rather than Student's t-test (Figures 1B, 1D, 1E, 1F).

All figure legends lack statistical test details.

Thanks for your advisable comment. All figure legends added statistical test details in statistical analysis.Minor comments:In all graphs, individual blastomere expression levels should be represented as boxwhisker/bar/scatter/violin plots since the comparison is groups rather than time points (i.e. symbols should not be connected with a line in Figures 1B, 1D, 1F-G, 1I, S1D, S1F).

Thanks for your advisable comment. Each colored line represents a single cell, and the dots of the same color represent the blastomere of the same cell. Thus, we use a line representation individual blastomere.

For all fluorescent images, having two representative images may be confusing for the reader. Figures may be improved by just including one representative image for each stage/treatment (Figures 1F, 1I, S1F, 3A, 3D, 4E, 4G).

Thanks for your advisable comment. The figures just including one representative image for each stage in our revised version. In addition, two representative images from each group were shown for each treatment (Figures 3A, 3D, 4E, 4G).

The manuscript would be improved with thorough grammar and typo editing.For example:(1) Lines 18, 73, the wording is confusing, consider: "knockdown of Hspa2 in one of the two-cell blastomeres biased its progeny towards the trophectoderm lineage.".(2) Line 23, overstatement. Consider: "we demonstrated that HSPA2 levels correlate with ICMassociated genes and that it interacts with the CARM1.".(3) Line 25 confusing wording, "via the execution of commitment and differentiation phases.".(4) Line 37, replace "that" with "of;" replace "cell-fate decisions" with "cell-fate decision".(5) Line 40: needs space before (CARM1).(6) Line 43: the wording is confusing, consider "can result in higher expression levels of".(7) Line 45: wording, consider "Recent [studies have] further suggested".(8) Line 70: plurality, consider "analyzed gene expression pattern".(9) Line 73 typo: "prevents its".(10) Line 76-77 wording, consider "Hspa2 expression patterns can bias cell fate in the mouse embryo".(11) Line 276: remove "in whole embryos," since MII eggs are not embryos.(12) Line 617 "There" should be "Three".(13) Axis label in Fig 3b "Totle" should be "Total".(14) Lines 417, 419 missing spaces.(15) Line 448 missing word, "interfering [with] the cell cycle".(16) Line 462 incorrect word, "[a]polar cells being specified as ICM".(17) Line 469 incorrect plural, "cell differentiation".

Thanks for your advisable comment. We revised the whole manuscript carefully according to the reviewers' suggestions.